# Assessing equity of the spatial distribution of primary health care facilities in Fuzhou City, China: A comprehensive method

**Ninglong You** *

College of Landscape Architecture, Fujian Agriculture and Forestry University, Fuzhou, Fujian, China

* 672777468@qq.com

## Abstract

**Data Availability Statement:** All data files are available from the figshare database (https://figshare.com/articles/dataset/data/17111453).

**Funding:** The author(s) received no specific funding for this work.

### Background

It is well known that equity assessment of the spatial distribution of primary health care facilities (PHCFs) is significant to optimize the allocation of health care resources and enhance the ability to react to public health emergencies, yet there is much discussion about the poor effectiveness of assessment for many cities due to the limitations of the insufficient ability of traditional data to reflect residents' demands, etc. In many cities where PHCFs are inequity distributed, this is urgently needed for the government.

### Methods

Using Fuzhou City, China as a case study, we propose a comprehensive method for assessing the equity that consisting of two frameworks based on the Geographic Information System. The first framework is assessing resident demand based on daily demand and potential demand, and the second is assessing PHCFs supply based on road impedance. This method combines an index system based on multi-source data and a spatial matching analysis between resident demand and PHCFs supply based on the supply-demand disparity index.

### Results

The demand degree for the PHCFs and the supply degree of accessing the PHCFs of different residential areas differ to great extents. The equity of the spatial distribution of PHCFs shows well overall, but there is still a lack of equity in local areas. The number of the residential areas with the lowest equity, insufficient supply and oversupply accounted for 10.31% and 39.00% respectively; the former is mainly gathered in a concentrated form at the edge and in a scattered form inside, and the latter is distributed in the interior mostly in a concentrated manner.

**Competing interests:** The authors have declared that no competing interests exist.

## Conclusion

The key findings highlight several aspects of improving the layout of PHCFs such as strengthen an in-depth analysis on residents' demand. This study contributes to a more accurate equity assessment, and further improves the layout of health care facilities.

## Introduction

Urban health care facilities offer the necessary resources and services to safeguard the health of residents and improve their livelihoods. The rationality of their spatial distribution is of great significance to the equitable allocation and effective utilization of health care resources, and it is essential to realize the national objective of "advancing the equalization of basic public services" in China [1, 2]. Primary health care facilities (PHCFs) are not only the core components of the health care service system but also a necessary condition for the promotion of its services. The rational spatial distribution of PHCFs for the cities ensures that all social groups, especially the elderly and other vulnerable groups, are more accessible to the essential health care services or even rescue service [3], so as to improve the health optimization goals of urban communities and the ability to react to public health emergencies [4].

The equity of facilities' spatial distribution not only reflects the difficulty of accessibility of services for residents in space [5], but also emphasizes the actual demands of required social groups in different regions from the view of the balance of supply and demand [6]. When the supply and demand of services match each other, the spatial distribution of the facility is considered to be fair for the society and its residents [7]. This is also the concept extension of equity in the study of the spatial distribution of PHCFs. Through the equity assessment of the current spatial distribution, this study identifies the areas where the supply and demand of the PHCFs do not match. It is theoretically and practically significant to the government for constructing a livable urban environment with equitable distribution of health care resources. Therefore, how to rationally carry out equity assessment of the spatial distribution of PHCFs has become a key topic among government departments and scholars.

Many cities use relevant studies to better assess the equity of basic health care facilities and focus on two aspects. On the one hand, the actual benefit from the angle of characteristics of the served groups (e.g., vulnerable or special social groups classified by individual attributes such as age, gender, income, education level, family structure, and living environment) is discussed to evaluate the equity of residents' access to needed facilities and services. The measurement methods include Gini coefficient, analysis of variance, construction of indices, spatial autocorrelation techniques and regression analysis, multilevel modeling, and multi-criteria decision making [8–13]. On the other hand, spatial accessibility measurements are adopted to analyze the pros and cons of the geographic location. The quantitative methods include the gravity model-based two-step floating catchment area method, potential model-based spatial equity analysis, spatial methods based on the access-based model, and new gravity-based spatial equilibrium evaluation [1, 14–16].

However, the results of equity assessment are still not accurate enough and the effect is poor for many cities, due to the limitations in the existing methods [17–19]. In previous studies, the most used basic units for assessing equity of the spatial distribution of health care facilities are cities, census areas, and administrative regions (such as districts and streets), while less attention is paid to the relatively minor spatial units, such as communities and residential areas [20–22]. One of the main reasons lies in that there is a lack of data on the characteristics

of the population distribution. The traditional data (such as China census data) is mostly obtained based on statistical units like streets or larger areas. In fact, in the construction of the equity assessment method, the spatial unit carrying the residents is usually abstracted into a point to simplify the impact factors on the evaluation. However, the smaller the spatial unit is, the more effective it can reflect the actual demands of facility served groups and the differences of accessibility, and a more accurate evaluation unit is more effective in guiding local spatial optimization [23]. It is necessary to further improve the spatial unit in assessing equity.

Meanwhile, residents' demand is a key factor to evaluate the spatial distribution of current facilities and allocation optimization. Many studies primarily measure residents' demand via the statistics of the facility served group population or by the means of sampling surveys of residents' demand [24–27]. However, these methods usually cannot reflect the actual demand completely due to the difficulty in obtaining data of the specific group, the randomness of the respondents, and the superficiality of the investigation contents. This often overlooks some potential demanding differences and the corresponding services for potential users [28]. For instance, as one of the main users of PHCFs, the elderly in the two areas have different demands due to the different activity features even they are of the same population. Overlooking this point may lead to differences in evaluation results. The application of the multi-source data combining traditional data, Internet data, and urban thermal data offers a way to overcome this limitation [29]. Last, the calculation of travel distance and time between the facilities and the served objects mainly depends on the network analysis. Yet there are not many discussions on the relevant factors (such as road accessibility and road attraction) that impact the actual traffic speed or time in origin-destination measurements, and there is a lack of analysis on the road attributes and real environment [30]. It is necessary to deepen equity assessment of the spatial distribution of PHCFs.

Addressing these limitations and the need for optimizing existing methods, this study uses Fuzhou City, China as a case study, and establishes a comprehensive method for assessing equity of the spatial distribution of PHCFs. The comprehensive method consists of two frameworks. The first framework is assessing resident demand based on daily demand and potential demand, and the second is assessing PHCFs supply based on road impedance. Residential areas are used as the basic spatial units carrying residents' demand, and an index system is constructed to assess residents' demand and PHCFs' supply based on multi-source data. Finally, a spatial matching between resident demand and PHCFs supply is analyzed by introducing a supply-demand disparity index based on the results of two frameworks. This study contributes to providing a more accurate evaluation of the spatial distribution of PHCFs and a more accurate equity assessment of it. It will help the governments in China and other countries to further improve the layout of health care facilities, enhance the urban community health and improve the ability to react to public health emergencies, to build a more livable urban environment.

## Methods

### Study area

As the capital city of Fujian Province, China, Fuzhou is one of the central cities in the Economic Zone on the Western Coast of the Taiwan Straits with a long history. The core area of Fuzhou City is selected as the case study, which is the main built-up area of the main urban area with an area of 24.57 square kilometers and a registered population of approximately 850,000 till the end of the year 2010. The dense population and concentrated constructions, lots of old residential areas, and the clustering of elderly and other vulnerable groups make its basic health care services need to be improved. As the prerequisite for service improvement, a

more accurate equity assessment of the spatial distribution of PHCFs is urgently needed. Meanwhile, the residential area is the basic composition of residents' settlements in Chinese cities, and the basic unit of real estate development projects. This study takes the PHCFs and residential areas within the core area as the study object and discusses the equity of services in the distribution of residential areas in order to determine whether residents can access services effectively.

## Data source

This study is carried out in accordance with the construction of a multi-source dataset. The types of data include: first, spatial data: layers of the residential area and building (including geographic locations and boundaries), and spatial point data of commercial enterprises and medical institutions. There are three levels of medical institutions in China. The first level is the lowest PHCFs providing basic health care services to the residents in surrounding residential areas. The second level is to provide larger and more medical care to more residents. The third level is the highest level, which can provide comprehensive medical care to the residents of the whole city and even beyond. Based on research purposes, the PHCFs (including community health care service centers (stations) and health centers (clinics)) are used as the object of study. Other levels of medical institutions (general hospitals) are grouped as one of the factors to analyze the current situation of accessing health care services for residential areas. According to statistics, there are 718 residential areas and 11,027 buildings. This data was obtained from the Baidu Map API (http://api.map.baidu.com/) and adjusted according to the OpenStreetMap (OSM) data (http://www.openstreetmap.org/). 11,652 commercial enterprises were obtained from the Baidu Map API. 78 medical institutions (57 PHCFs and 21 general hospitals) were obtained from the Baidu Map API and optimized by official data from the Urban Planning Authorities.

Second, road data: the distribution, grade, and width of roads. They were obtained from the current road analysis of the "Master Plan of Fuzhou City (2011–2020)" and Baidu Map. Population data: the floating population and elderly population at the street level. These data were obtained from the statistical data in the "Population Census Data of Fuzhou City". Property data of residential area: the number of inhabitants, time of completion, greening rate, and housing price. They were obtained from Fuzhou real estates related websites, such as http://m.jiwu.com/fz/ and https://fz.anjuke.com/. Urban thermal data: residents' thermal data at 10:00 a.m. on the weekday (May 25, 2020) from different residential areas. There was a total of 9,457 thermal points, which were obtained from the service window of location big data of Tencent Series Products (https://heat.qq.com/index.php).

## Comprehensive method

Considering the research purpose and the assessments of resident demand and PHCFs supply, this study establishes two frameworks to assess the equity of the spatial distribution of PHCFs in a comprehensive method. Framework 1 is assessing resident demand based on two indices of daily demand and potential demand with a total of seven evaluation factors. The analysis procedures consist of determining the weight of evaluation factors by the Analytic Hierarchy Process (AHP); modifying and screening the evaluation factors through normalization processing and multiple linear regression analysis; calculating the demand value of each residential area using the weighted sum of evaluation factors; and classifying the demand values and the demand degree of each residential area by K-means clustering analysis (KCA). Framework 2 is assessing PHCFs supply based on the road impedance with a total of three evaluation factors. The procedures consist of determining the weight of evaluation factors by the AHP;

calculating the impedance value for each road by the weighted sum of evaluation factors; classifying the road impedance values by KCA to distinguish the actual walking speed; scoping the service range by combining the actual walking speed and walking time, which is used as the basis for classifying the supply degree. Finally, the differences between the demand degree and the supply degree of residential areas are analyzed by introducing a supply-demand disparity index (SDI). The residential areas with large supply-demand differences (i.e., mismatch) are identified, which are used as the key areas for prioritizing improve the layout of PHCFs (Fig 1).

**Index system.** The index system is established based on the previous studies, and types of multi-source data, to analyze the actual supply and demand. It is divided into resident demand and road impedance. Resident demand indicates the degree of residents' demand for PHCFs due to the characteristics of residential areas, and road impedance indicates the degree of spatial obstruction in providing PHCFs services through roads due to their features [22]. Among them, the resident demand consists of daily demand and potential demand, which includes four evaluation factors: the number of inhabitants (NI), the time of completion (TC), the greening rate (GR), and the distance to the nearest general hospital (DH); and three evaluation factors: the vitality of the elderly (VE), the economic vitality (EV), and the floating population (FP), respectively. The road impedance includes three evaluation factors: the road grade (RG), the road width (RW), and the height of buildings along the road (HB).

For daily demand: (1) Population scale is one of the determinants of the allocation amount of PHCFs. It is usually considered that the more population it has, the more demands for PHCFs it requires. In this study, the NI in a residential area is used to represent the population scale. (2) Perfecting basic medical care in old residential areas has been one of the core elements to improve China's urbanization through urban governance in recent years. In fact, the population usually deviates from the number of facilities in the urban area [31]. It is necessary to analyze the attributes of the residential area beside the population when evaluating the resident demand. In general, the older the residential area, the less allocation of the PHCFs and the greater the demand. The TC of the residential area is therefore a factor affecting resident demand. (3) There are situations of large differences in the allocation of facilities among the residential areas of similar ages. The GR is used as a supplement to the evaluation of TC. As one of the important supporting facilities, the higher the GR is, the better the overall configuration of the facilities it reflects. (4) PHCFs can be replaced by general hospitals in providing services. When the residential area is near the general hospital, the demands for PHCFs will be reduced. The DH is positively correlated with the resident demand.

For potential demand: (1) The elderly are the main service objects of the facilities with treatment, sanitation, and health care [32]. The more the elderly, especially those with independent activity ability, there are, the more demands for the PHCFs there needs. However, there is a lack of statistics on the elderly population at the residential area level. This study adopts the VE to describe the elderly population with independent activity ability [33], which is reflected by the residential thermal data of the residential area during weekday hours when the active residents are mainly non-laborers like the elderly. (2) The convenient commercial facilities in the residential area meet the daily demands of residents due to its small service radius, especially for the elderly [34]. The more they are, the more elderly there will be to some extent. This study adopts the EV as a factor reflected by the number of convenient commercial facilities (such as small supermarkets and convenience stores). (3) It is an orientation to promote social integration and improve the life quality of residents through the efforts in solving the practical problems faced by the floating population and improving the service system of medical care. The FP is adopted as a factor. Firstly, the amount of FP at the street level is allocated to each residential area based on its residential construction area. Then, according to the current

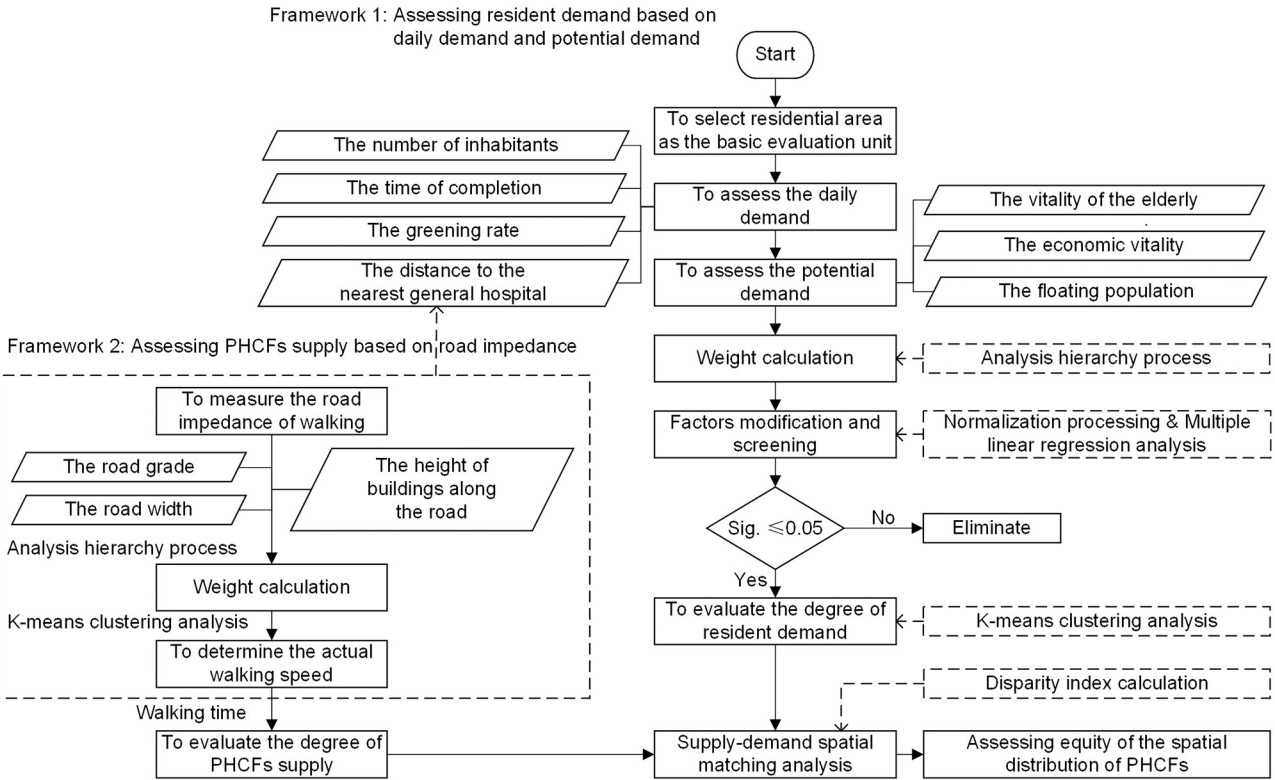

**Fig 1. The comprehensive method for assessing equity and analysis procedures.**

housing price of residential areas, the amount of FP is revised. Studies show that the type of residential area classified by housing price is somehow influential on the accessibility to public services [35].

For road impedance: (1) The walkability of residents especially the elderly to access services and the impacts from the road environment is significant for evaluating the distribution of facilities [30]. The RG is proportional to the traffic capacity, i.e., the higher grade the roads connecting to the residential areas have, the better the accessibility is, and the more services provided by the PHCFs the residents can get. (2) Since there are many regions in the old city where the road grade is not proportional to its width, the RW is adopted as another representation of road geometric features. (3) The height of the surrounding buildings, which is one of the environmental conditions along the road. It has an impact on accessibility (especially on foot) as well. The higher the buildings are, the harder it is to create a pleasant pedestrian environment and the less attractive it is for population aggregation [36]. In this study, the HB is adopted as the factor, i.e., average building height within a 30-meter buffering zone along the road.

**Evaluation framework.** The AHP is applied to calculate the weight of each evaluation factor in the assistant software of YAAHP. Judgment matrices are established respectively for the evaluation factors of resident demand and road impedance. Combining the interviews with experts and residents, the weight of the factor is determined through comparisons among the evaluation factors. The consistency ratio of weight results is then judged to see whether it is less than 0.1 to meet the consistency requirements.

The IBM SPSS Statistics 23 is used to normalize the values of evaluation factors and the specific data is converted into decimals between 0 and 1. The NI, DH, VE, EV, and FP are

positively correlated with the resident demand, while the TC and GR are negatively correlated with the resident demand. The HB is positively correlated with the road impedance, and the RG and RW are negatively correlated with road impedance. Therefore, the factors of positive correlation and negative correlation are respectively converted by Eqs (1) and (2).

$$x_{ij\_nor} = \frac{x_{ij} - min\left\{x_{1j}, x_{2j}, \ldots, x_{nj}\right\}}{max\left\{x_{1j}, x_{2j}, \ldots, x_{nj}\right\} - min\left\{x_{1j}, x_{2j}, \ldots, x_{nj}\right\}} \tag{1}$$

$$x_{ij\_nor} = \frac{max\left\{x_{1j}, x_{2j}, \ldots, x_{nj}\right\} - x_{ij}}{max\left\{x_{1j}, x_{2j}, \ldots, x_{nj}\right\} - min\left\{x_{1j}, x_{2j}, \ldots, x_{nj}\right\}} \tag{2}$$

Where $x_{ij}$ is the value of the jth evaluation factor for the ith residential area; $n$ is the number of residential areas; $x_{ij\_nor}$ is the normalized value of the jth evaluation factor for the ith residential area. For the road impedance, $x_{ij}$ is the value of the jth evaluation factor for the ith road; $n$ is the number of roads; $x_{ij\_nor}$ is the normalized value of the jth evaluation factor for the ith road.

The multiple linear regression analysis on the resident demand and evaluation factors is applied to optimize the index system. Scatter analysis is performed for each factor. The results reveal that the data of TC and GR are normally distributed, the rest of the factors are non-normally distributed. According to its image characteristics, the data of non-normally distributed factors are function calculated and transformed into a normal distribution. Through several experiments, the NI, VE, and FP are selected by function $sin\sqrt[4]{x}$, DH is selected by function $\sqrt{\log_{10}(x+1)}$, and EV is selected by function $\log_{10} x$.

Based on the results, the multiple linear regression analysis is performed on the converted factor data and the weight calculation-based evaluation results of the resident demand (Table 1). Overall, the determination coefficient $R^2$ is 0.887 with good fitting effects of the model. However, in terms of Sig. coefficient, the significance of the EV is not high, so it is removed to raise the model fit.

The NI, TC, GR, DH, VE, FP, are determined to measure the resident demand after being tested by regression analysis. Meanwhile, the demand value of each residential area is calculated according to the data of the evaluation factor and Eq (3).

$$RD = Wi_{NI}sin\sqrt[4]{x_{NI\_nor}} + Wi_{TC}x_{TC\_nor} + Wi_{GR}x_{GR\_nor} + Wi_{DH}\sqrt{\log_{10}(x_{DH\_nor} + 1)} + Wi_{VE}sin\sqrt[4]{x_{VE\_nor}} + Wi_{FP}sin\sqrt[4]{x_{FP\_nor}} \tag{3}$$

Where $RD$ is the value of resident demand; $Wi$ is the weight of each evaluation factor; $x_{NI\_nor}$, etc. are the normalized data of evaluation factors.

The KCA is performed on the weighted-summed demand values of each residential area, and the results are classified into four categories, with the clustering central values at 0.4119, 0.5121, 0.5923, and 0.6870. It indicates the four degrees of resident demand from low to high.

Meanwhile, the KCA is performed on the impedance value of each road as well, and the results are classified into three categories, with the clustering central values at 0.2844, 0.4513, and 0.5902. It indicates the degrees of road impedance from low to high, that is, the supply degrees of PHCFs from high to low. Corresponding walking speed is further assigned according to the degrees of road impedance. The lower the impedance is, the faster the walking speed is, i.e., the walking speed of 0.9 m/s, 0.8 m/s, and 0.7 m/s are assigned with the clustering central values respectively [37]. Finally, combining with the actual road condition, the facility

**Table 1. Results of multiple linear regression analysis on the evaluation factors of resident demand.**

| Model | | Unstandardized Coefficients | | Standardized Coefficients | t | Sig. |
|---|---|---|---|---|---|---|
| | | B | Std. Error | Beta | | |
| 1 | (Constant) | -0.218 | 0.008 | | -25.725 | 0.000 |
| | lg_NI | 0.089 | 0.011 | 0.111 | 7.741 | 0.000 |
| | lg_TC | 0.039 | 0.007 | 0.079 | 5.600 | 0.000 |
| | lg_GR | 0.034 | 0.009 | 0.055 | 3.970 | 0.000 |
| | lg_DH | 0.736 | 0.011 | 0.846 | 65.950 | 0.000 |
| | lg_VE | 0.051 | 0.007 | 0.108 | 6.898 | 0.000 |
| | lg_EV | 0.000 | 0.001 | -0.005 | -0.360 | 0.719 |
| | lg_FP | 0.152 | 0.011 | 0.223 | 14.149 | 0.000 |

[a]. Dependent Variable: Resident demand.

Note: Sig. is the p-value of the t-test, which is generally considered significant if it is less than 0.05.

service scopes of 0–5 minutes, 5–10 minutes, 10–15 minutes, and over 15 minutes are formed to express the four supply degrees of PHCFs from high to low [38].

**Supply-demand spatial matching analysis.** Based on the classification of demand degree and supply degree, the four levels of demand degree and supply degree from low to high are assigned as 1, 2, 3, and 4, respectively. For the analysis of the spatial matching between resident demand and PHCFs supply, the Spearman Rank Correlation Coefficient is adopted to judge the overall matching degree of resident demand and PHCFs supply. Then, an SDI is introduced to exam the local spatial mismatches. There have already been applications of the SDI in the evaluation of the supply-demand differences of various facilities. The calculation results indicate the contributions of each evaluation unit to the overall differences, i.e., the larger the absolute value is, the lower the matching degree of supply and demand is, which also indicates the lower equity of accessing PHCFs of the corresponding residential area. The SDI can be calculated by Eqs (4) and (5) [39, 40].

$$SDI_{\_area} = \frac{\sum_1^n \left| \frac{dem_{\_i}}{dem_{\_area}} - \frac{sup_{\_i}}{sup_{\_area}} \right|}{2} \tag{4}$$

$$SDI_{\_i} = \frac{\frac{sup_{\_i}}{sup_{\_area}} - \frac{dem_{\_i}}{dem_{\_area}}}{2SDI_{\_area}} \tag{5}$$

Where $SDI_{\_area}$ is the supply-demand disparity index of the overall study area and its result is between 0 and 1; $dem_{\_i}$ is the degree of resident demand of the ith residential area and its value is 1, 2, 3, or 4; $dem_{\_area}$ is the total value of all residential areas' demand degree in the study area; $sup_{\_i}$ is the degree of facility supply of ith residential area and its value is 1, 2, 3, or 4; $sup_{\_area}$ is the total value of all residential areas' supply degree in the study area; $n$ is the number of residential areas; $SDI_{\_i}$ is the value of supply-demand disparity index of the ith residential area.

## Results

### Normalized results of evaluation factors

Through the normal distribution test, functions are established for the non-normally distributed evaluation factors of resident demand after normalization processing. According to the

results of multiple linear regression analysis, the NI, TC, GR, DH, VE, and FP are determined to be the evaluation factors to evaluate resident demand. The normalized values of resident demand and road impedance evaluation factors are calculated by Eqs (1) and (2). Due to the NI, DH, VE, and FP are positively correlated with the resident demand, the larger the value of their calculations, the more demands it requires. On the contrary, the TC and GR are negatively correlated with the resident demand, the larger the value is, the less the resident demand is. In a similar way, among the three factors to evaluate road impedance, the HB is positively correlated with the road impedance. The larger the value of the road, the higher the HB is, hence the greater the impedance is; while the RG and RW are negatively correlated with the road impedance, the larger the value is, the lower the grade is and the narrower the width is, hence the greater the impedance is. The greater the impedance is, the harder it is for residents to obtain services.

## Weighted results of evaluation factors

Judgment matrices are established for evaluation factors by the AHP to determine the weight. The results indicate that the consistency ratio of the judgment matrices for the resident demand is 0.0066 (less than 0.1), which satisfies the consistency requirement. The weights for the factors of NI, TC, GR, DH, DH, and FP are 0.1310, 0.0736, 0.0736, 0.3631, 0.2276, and 0.1310. The consistency ratio of the judgment matrices for the road impedance is 0.0088, which also meets the consistency requirement. The weights for the factors of RG, RW, and HB are 0.1634, 0.2970, and 0.5396. The larger the weight is, the greater impact the factors have on the evaluation indices.

## Resident demand

After determining the normalized values of each factor and corresponding weight, the resident demand of each residential area is calculated by Eq (3). The larger the value of resident demand of the residential area is, the more demands it requires, and vice versa. Overall, the degree of resident demand in different residential areas varies significantly.

Then, this study further adopts a KCA on the values of resident demand to evaluate the degree of resident demand. Meanwhile, each residential area is converted into a centroid to help overlay analysis with PHCFs supply. Residential areas are separated into four categories based on the KCA results of demand values, and a scale of 1 to 4 indicates that the degree of resident demand increases gradually. The points with a value at 4 indicate the highest degree of resident demand in this category, with 196 residential areas, accounting for 27.22% of the total number of residential areas. The obvious characteristics lie in the large population, especially the elderly and floating population. Besides, the distance to general hospitals is relatively far, and the construction age is relatively old which causes a lack of facilities. Therefore, the demand for PHCFs in these residential areas is the highest. The value at 3 and 2 indicate the medium degree, with 138 (19.17%) and 297 (41.25%) residential areas, respectively. The value at 1 indicates that the degree is the lowest, a total of 87 (12.08%) residential areas.

## PHCFs supply

Road impedance (i.e., the difficulty degree of the service provision for residents by roads) is adopted to represent the situation of PHCFs supply. The impedance evaluation of the current road is obtained by the weighted sum of the three factors: RG, RW, and HB. The larger the value of the road is, the higher the impedance of the road is and the more difficult it is to provide service for residents.

According to the principle that the larger the road impedance is, the slower the walking speed of residents is, the results of KCA are assigned with different walking speeds. The results of road clustering are divided into three categories of 1, 2, 3 and the degree of impedance increases gradually. The length of those three categories of roads accounts for 30.38%, 55.02%, and 14.60% of the total length of the roads. Finally, four service scopes are formed according to the different walking speeds assigned to different impedance roads. The supply degree of residential areas within the ranges of 0–5 minutes, 5–10 minutes, 10–15 minutes, and over 15 minutes is assigned a value of 4, 3, 2, and 1 respectively, which indicates that their supply of service reduces gradually. Residential areas within the 0–5 minutes are considered to have the best PHCFs supply, 194 (26.94%) in total, where the proportion of 4 types of residential areas with demand from high to low are 28.87%, 18.56%, 44.33%, and 8.25%. There are 285 (39.58%) residential areas within the 5–10 minutes, where the proportions of 4 types of residential areas with demand from high to low are 29.12%, 18.95%, 41.75%, and 10.18%. Distance within 10 minutes walking is generally regarded as the optimal service scope of facilities at the residential area level [1]. There are 160 (22.22%) residential areas within the 10–15 minutes, where the proportion of 4 types of residential areas with demand from high to low are 26.25%, 19.38%, 35.63%, and 18.75%. However, there are 81 (11.25%) residential areas within more than 15 minutes, which means that PHCFs is difficult to meet the basic health care demands for the residents. The proportion of 4 types of residential areas with demand from high to low are 18.52%, 20.99%, 45.68%, and 14.81%.

## Equity of the spatial distribution of PHCFs

The values of 1, 2, 3, and 4 are assigned to the resident demand and PHCFs supply respectively from low to high according to the classification of their degrees. Then the Spearman Rank Correlation Coefficient is applied to judge the overall matching between them. The result is 0.073*, indicating that the matching between them is significant at the level of 0.05. It reveals that the resident demand and the supply are matched overall with good equity.

Furthermore, the supply-demand situation of the PHCFs in local areas is deeply assessed through Eqs (4) and (5). This study classifies the results of SDI into five grades, less than or equal to -0.002000, -0.001999 to -0.001000, -0.000999 to 0.000000, 0.000001 to 0.001000, and greater than or equal to 0.001001, respectively. Negative values indicate that resident demand is higher than PHCFs supply, while positive values indicate that PHCFs supply is higher than resident demand. The greater the absolute values of the results of each residential area are, the greater the differences between supply and demand are, i.e., the lower the equity of the spatial distribution is.

The residential areas with values less than or equal to -0.002000 and greater than or equal to 0.001001 are the ones with the lowest equity, representing serious insufficient supply and oversupply respectively; the residential areas with values between -0.001999 to -0.001000 and 0.000001 to 0.001000 are the ones with average equity, indicating a certain degree of insufficient supply and oversupply; the rest with values from -0.000999 to 0.000000 are the ones with better equity. The number of residential areas with five grades of SDI from low to high are 74, 150, 178, 36, and 280 (i.e., 10.31%, 20.89%, 24.79%, 5.01%, and 39.00%) (Fig 2a). In other words, the proportions of residential areas with the lowest, average, and better equity account for 49.31%, 25.90%, and 24.79% of the total number of residential areas. Further analysis of the residential areas with the lowest equity reveals that for the situation that the demand is higher than the supply, the numbers of residential areas within the PHCFs service scope of 0–5 minutes, 5–10 minutes, 10–15 minutes, and over 15 minutes are 1, 0, 42, and 31 (i.e., 1.35%, 0.00%, 56.76%, and 57.11%), which means that most residential areas are within 10–15 minutes' or

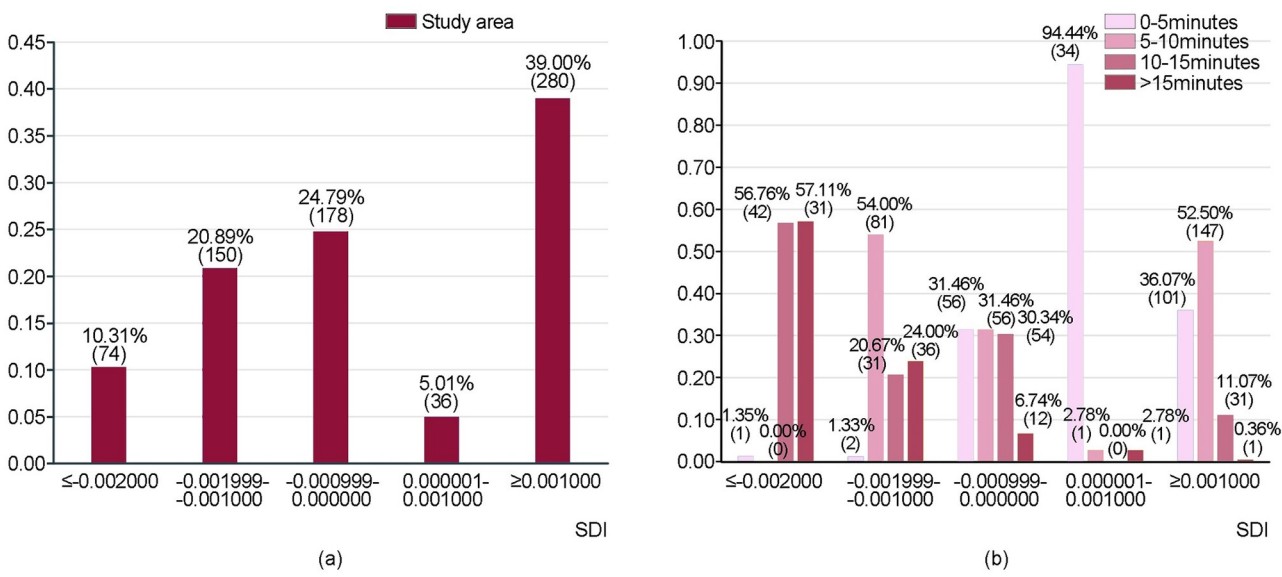

**Fig 2. Equity results of the PHCFs.** (**a**) The number and percentage of the residential area with each SDI. (**b**) The number and percentage of the residential area with each SDI in different service scopes.

over 15 minutes' service scope. While for the situation that the supply is greater than the demand, the numbers of the residential areas within the four service scopes mentioned above are 101, 147, 31, and 1, (i.e., 36.07%, 52.50%, 11.07%, and 0.36%), which means that most residential areas are within 0–5 minutes' or 5–10 minutes' service scope (Fig 2b).

Then, the spatial distribution of residential areas with various SDI is analyzed. In the case of less than 0, the smaller the value is, the lower the equity is, indicating the insufficient supply. In the case of greater than 0, the larger the value is, the lower the equity is, indicating the oversupply. Spatial interpolation analysis on the SDI is conducted through the inverse distance weighted method to summarize its overall situation. The residential areas with the lowest equity and with a serious shortage of PHCFs supply are mainly gathered in a concentrated form at the edge of the core area, such as Hongxing Community, Binjiang Community, and Donghu Community, while these residential areas are showed a scattered form inside. The residential areas located in these communities are relatively large in scale with far distances to general hospitals, and the number of PHCFs is relatively small. However, the residential areas with the lowest equity and greater oversupply are mainly distributed in the interior of the study area mostly in a more concentrated manner, such as Shanghaixinyuan Community, Jindou Community, and Dadao Community. The residential areas located in these communities are relatively small in scale with close distances to general hospitals, and the number of PHCFs is relatively large and concentrated. It is revealed that the equity of spatial distribution of PHCFs varies greatly, particularly in the areas with relatively low equity.

## Discussion

The purpose of the geographical equity assessment of PHCFs is to ensure that urban residents can have undifferentiated access to the services they need, especially the special groups with higher needs for PHCFs [10]. And the construction of a comprehensive method aims to achieve this assessment more accurately. Through the construction of an index system based on multi-source data and spatial matching analysis based on the supply-demand disparity

index, the demand degree and supply degree of PHCFs and their matching relationship are measured. For many cities, residents' effective access to health care resources is still inadequate, this study provides a new method to identify whether its spatial distribution is fair or not, and it is suitable for micro-level analysis.

For a long time, the effect of equity assessment of PHCFs' spatial distribution by the government is poor [17]. The proposed two evaluation frameworks ensure a reasonable assessment of degrees of demand and supply. First, the assessment of the demand degree is not only based on the population scale, but also takes indices that have an impact on the number of inhabitants with high demand for PHCFs and residents' opportunities to use PHCFs (i.e. daily demand) as the main basis for the assessment, and indices that have an impact on residents' demand for using PHCFs (i.e. potential demand). Second, the assessment of the supply degree is mainly based on the degree of difficulty for residents to obtain PHCFs services (i.e. road impedance). The matching degree (i.e. equity) between those two is analyzed by the supply-demand disparity index, which provides a method to measure equity of the spatial distribution of PHCFs in residential areas and fully considers the demand and supply of health care resources.

Many current health care resource allocation policies in Chinese cities are made rarely from the perspective of space or geography [41], which leads to great differences in the equity of residents to obtain health care resources in different areas within cities. This is also one of the main advantages of using the comprehensive method to identify areas with poor equity in order to optimize health care resource allocation. In the case study of Fuzhou City, residential areas with the less equitable spatial distribution of PHCFs were identified. Specifically, the spatial distribution of PHCFs is reasonable overall, but there is still a lack of equity in local areas. Combining the previous studies on the analysis of the current situation of urban health care resource allocation in Chinese cities [42, 43] and the findings of this study, to explore the potential reasons for this phenomenon. There is a lack of in-depth analysis on the current residents' demand in different residential areas, especially on the population characteristics with higher demand for PHCFs; the consideration on the equalization of the overall layout of PHCFs in spatial planning is not enough detail, leading to insufficient supply or oversupply in local areas due to excessive concentration or dispersion; there is an inadequate overall plan with higher-level medical institutions in the PHCFs planning to reflect the idea of gradual deepening and mutual complementing; the attention to the accessibility of roads is insufficient, resulting in difficulty in obtaining services.

In the future PHCFs layout improvement, it is necessary to strengthen the investigations on the demands of different social groups to better satisfy the actual demand in the local areas. Then, the numbers of PHCFs in areas with oversupply shall be controlled, and functions of partial existing PHCFs shall take the replacement into consideration to improve the utilization rate. The allocation of PHCFs should be increased in the means of construction and replacement to improve the accessibility to services for the residents in areas with insufficient supply. Thirdly, for the site selection of PHCFs, it is necessary to avoid choosing the areas too close to the existing medical institutions with the same or higher level, to reduce the waste of health care resources. Finally, for the reconstruction of the urban environment, low-grade roads such as branch roads shall be added, and the existing road conditions shall be considered to be improved. This is one of the ways to improve the equity of health care services in terms of non-facilities themselves.

## Conclusions

This study focuses on how to assess the equity of the spatial distribution of PHCFs more accurately in the context of the need to improve the effectiveness of assessment for many cities. A

comprehensive method consisting of two frameworks is proposed. The first framework is assessing resident demand based on daily demand and potential demand, and the second is assessing PHCFs supply based on road impedance. This method will contribute to providing a more accurate equity assessment of the spatial distribution of PHCFs, which helps the governments in China and other countries to further improve the layout of health care facilities and the ability to react to public health emergencies.

There are several key findings: (1) According to the evaluation results of resident demand and PHCFs supply, the demand degree for the PHCFs and the supply degree of accessing the PHCFs of different residential areas differ to great extents. (2) The equity of the spatial distribution of PHCFs shows well overall, but there is still a lack of equity in local areas due to the mismatch between resident demand and PHCFs supply. The proportions of the residential areas with the lowest equity and insufficient supply and the residential areas with the lowest equity and oversupply account for 10.31% and 39.00%, respectively. (3) The residential areas with a serious supply shortage are mainly gathered in a concentrated form at the edge of the study area and in a scattered form inside. The residential areas with serious oversupply are mainly distributed in the interior of the study area mostly in a more concentrated manner. These findings are significant to Fuzhou City government for constructing a livable urban environment with the equitable spatial distribution of health care resources.

This study was limited to does not take the impact of the capacity (or service ability) of PHCFs on the equity assessment into account. Meanwhile, due to the lack of data, the consideration of resident demand, road impedance, and other related factors (such as social factors affecting residents' preference and guidance factors affecting road access) are still relatively insufficient, which may cause some inaccuracy in the results. These deficiencies will be constantly improved in future research.

## Acknowledgments

This research is supported by the data collected by the Urban Planning Authorities of Fuzhou City.

## Author Contributions

**Conceptualization:** Ninglong You.

**Data curation:** Ninglong You.

**Formal analysis:** Ninglong You.

**Methodology:** Ninglong You.

**Software:** Ninglong You.

**Visualization:** Ninglong You.

**Writing – original draft:** Ninglong You.

**Writing – review & editing:** Ninglong You.

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
