## [Decision Letter · Decision Letter 0]

12 Oct 2021

PONE-D-21-20230Assessing equity of the spatial distribution of primary health care facilities in Fuzhou City, China: A comprehensive methodPLOS ONE

Dear Dr. YOU,

Thank you for submitting your manuscript to PLOS ONE. After careful consideration, we feel that it has merit but does not fully meet PLOS ONE’s publication criteria as it currently stands. Therefore, we invite you to submit a revised version of the manuscript that addresses the points raised during the review process.

ACADEMIC EDITOR: Considering my own reading and reviewer suggestion, I am recommending a minor revision for this manuscript. Please address the minor comments made by reviewer and submit the revision. 

We look forward to receiving your revised manuscript.

Kind regards,

Srinivas Goli, Ph.D.

Academic Editor

PLOS ONE

Journal Requirements:

3. We note that Figure 1, 3, 4, 5 and 7 in your submission contain [map/satellite] images which may be copyrighted. All PLOS content is published under the Creative Commons Attribution License (CC BY 4.0), which means that the manuscript, images, and Supporting Information files will be freely available online, and any third party is permitted to access, download, copy, distribute, and use these materials in any way, even commercially, with proper attribution. For these reasons, we cannot publish previously copyrighted maps or satellite images created using proprietary data, such as Google software (Google Maps, Street View, and Earth). For more information, see our copyright guidelines: http://journals.plos.org/plosone/s/licenses-and-copyright.

a. You may seek permission from the original copyright holder of Figure 1, 3, 4, 5 and 7 to publish the content specifically under the CC BY 4.0 license.  

Additional Editor Comments:

Considering my own reading and reviewer suggestion, I am recommending a minor revision for this manuscript. Please address the minor comments made by reviewer and submit the revision.

Reviewers' comments:

Reviewer's Responses to Questions

**Comments to the Author**

1. Is the manuscript technically sound, and do the data support the conclusions?

Reviewer #1: Yes

2. Has the statistical analysis been performed appropriately and rigorously? 

Reviewer #1: Yes

3. Have the authors made all data underlying the findings in their manuscript fully available?

Reviewer #1: Yes

4. Is the manuscript presented in an intelligible fashion and written in standard English?

Reviewer #1: Yes

5. Review Comments to the Author

Reviewer #1: Dear Editor, Thank you for giving me the exciting opportunity to review the present manuscript for a prestigious journal.

In my opinion, the equity assessment of the spatial distribution of primary health care facilities is an interesting and important issue. A comprehensive method consisting of two frameworks is proposed. First framework is assessing resident demand based on two indices of daily demand and potential demand with a total of seven evaluation factors. Second framework is assessing primary health care facilities supply based on the road impedance with a total of three evaluation factors.

I think data and methods were chosen appropriately. A valuable result is finding that the equity of the spatial distribution of PHCFs shows well overall, but there is still a lack of equity in local areas due to the mismatch between resident demand and PHCFs supply. The paper itself is well structured and the study well conducted. I agree that this study would gain a lot if the impact of the service ability of primary health care facilities on the equity assessment were taken into account.

In my opinion this paper is a valuable contribution to the field. This paper might be useful for many researchers.

6. PLOS authors have the option to publish the peer review history of their article (what does this mean?). If published, this will include your full peer review and any attached files.

Reviewer #1: No

---

## [Author Response · Author response to Decision Letter 0]

6 Nov 2021

Response: We appreciate the time and efforts by the editor and reviewers in reviewing this manuscript. We have made modifications according to the comments of the editor and reviewers, and carried out the corrections one by one as suggested by the reviewers. All the changes have been highlighted in red in the revised manuscript. We hope that the revised version can meet the requirements of journal publication. 

Reviewers' comments:

(1) The paper itself is well structured and the study well conducted. I agree that this study would gain a lot if the impact of the service ability of primary health care facilities on the equity assessment were taken into account.

Response: Firstly, we appreciate your recognition of my research work and manuscript writing. Secondly, for the comment “I agree that this study would gain a lot if the impact of the service ability of primary health care facilities on the equity assessment were taken into account”, I have considered this and mentioned it as “limitations” in the last paragraph at the end of the manuscript. At this stage, the impact of the service ability of primary health care facilities on the equity assessment was not taken into account due to the lack of relevant data. We will take it as one of the important directions and optimize it in future research. Thank you very much for your constructive suggestions.

Response: Thank you very much, we have revised the styles of the manuscript body and title page according to the PLOS ONE style templates that you provided.

(2) Please ensure that you include a title page within your main document.

Response: Thank you very much, we have added the title page into the beginning of my manuscript.

(3) We note that Figure 1, 3, 4, 5 and 7 in your submission contain [map/satellite] images which may be copyrighted.

Response: Thank you very much, we have removed figures 1, 3, 4, 5 and 7 in the manuscript because it is difficult to obtain permission from the original copyright holder, and it has no great influence on the content expression of the manuscript. The changes of the corresponding part in the content expression have been highlighted in red in the revised manuscript.

(4) Please review your reference list to ensure that it is complete and correct.

Response: Thank you very much, we have reviewed my reference list and all the changes have been highlighted in red.

---

## [Decision Letter · Decision Letter 1]

18 Nov 2021

PONE-D-21-20230R1Assessing equity of the spatial distribution of primary health care facilities in Fuzhou City, China: A comprehensive methodPLOS ONE

Dear Dr. YOU,

Thank you for submitting your manuscript to PLOS ONE. After careful consideration, we feel that it has merit but does not fully meet PLOS ONE’s publication criteria as it currently stands. Therefore, we invite you to submit a revised version of the manuscript that addresses the points raised during the review process.

ACADEMIC EDITOR: Please insert comments here and delete this placeholder text when finished. Be sure to:Indicate which changes you require for acceptance versus which changes you recommendAddress any conflicts between the reviews so that it's clear which advice the authors should followProvide specific feedback from your evaluation of the manuscriptPlease ensure that your decision is justified on PLOS ONE’s publication criteria and not, for example, on novelty or perceived impact.

We look forward to receiving your revised manuscript.

Kind regards,

Srinivas Goli, Ph.D.

Academic Editor

PLOS ONE

Journal Requirements:

Additional Editor Comments (if provided):

Considering the reviewers suggestion, I am going with a decision of minor revision.

Reviewers' comments:

Reviewer's Responses to Questions

**Comments to the Author**

1. If the authors have adequately addressed your comments raised in a previous round of review and you feel that this manuscript is now acceptable for publication, you may indicate that here to bypass the “Comments to the Author” section, enter your conflict of interest statement in the “Confidential to Editor” section, and submit your "Accept" recommendation.

Reviewer #1: All comments have been addressed

2. Is the manuscript technically sound, and do the data support the conclusions?

Reviewer #1: Yes

3. Has the statistical analysis been performed appropriately and rigorously? 

Reviewer #1: Yes

4. Have the authors made all data underlying the findings in their manuscript fully available?

Reviewer #1: Yes

5. Is the manuscript presented in an intelligible fashion and written in standard English?

Reviewer #1: Yes

6. Review Comments to the Author

Reviewer #1: (No Response)

7. PLOS authors have the option to publish the peer review history of their article (what does this mean?). If published, this will include your full peer review and any attached files.

Reviewer #1: No

---

## [Author Response · Author response to Decision Letter 1]

21 Nov 2021

Dear Editor:

We appreciate the time and efforts of the editor and reviewers in the second review process of this manuscript. We have made responses for the comments of the editor, reviewers, and journal one by one. All the changes have been highlighted in red in the revised manuscript. We hope that the revised version can fully meet the requirements of journal publication.

Response to Reviewers' Comments:

Comments:

1. If the authors have adequately addressed your comments raised in a previous round of review and you feel that this manuscript is now acceptable for publication, you may indicate that here to bypass the “Comments to the Author” section, enter your conflict of interest statement in the “Confidential to Editor” section, and submit your "Accept" recommendation.

Reviewer #1: All comments have been addressed

2. Is the manuscript technically sound, and do the data support the conclusions?

Reviewer #1: Yes

3. Has the statistical analysis been performed appropriately and rigorously?

Reviewer #1: Yes

4. Have the authors made all data underlying the findings in their manuscript fully available?

Reviewer #1: Yes

5. Is the manuscript presented in an intelligible fashion and written in standard English?

Reviewer #1: Yes

6. Review Comments to the Author

Reviewer #1: (No Response)

Response: Thank you very much for your approval of my responses to your comments of the first review, and thank you for your recognition of my research work and other aspects as well.

Response to Additional Editor Comments:

Comments:

Considering the reviewers suggestion, I am going with a decision of minor revision.

Response: Thank you very much for your review of my manuscript, I have carefully revised it and made it meet PLOS ONE’s publication criteria.

Response to Journal Requirements:

Requirements: 

Response: Thank you very much, we have reviewed my reference list again. Three references were modified and the changes have been highlighted in red in the “Revised Manuscript with Track Changes”. They are as follows:

(1) 4. Farhang L, Bhatia R, Scully CC, Corburn J, Gaydos M, Malekafzali S. Creating tools for healthy development: Case study of San Francisco’s eastern neighborhoods community health impact assessment. J Public Health Man 2008; 14(3):255-65. doi: 10.1097/01.PHH.0000316484.72759.7b.

(2) 39. Liu HX, Remme RP, Hamel P, Nong HF, Ren H. Supply and demand assessment of urban recreation service and its implication for greenspace planning: A case study on Guangzhou. Landsc Urban Plan 2020; 203:103898. doi: 10.1016/j.landurbplan.2020.103898.

(3) 40. Lee G, Hong I. Measuring spatial accessibility in the context of spatial disparity between demand and supply of urban park service. Landsc Urban Plan 2013; 119:85-90. doi: 10.1016/j.landurbplan.2013.07.001.

---

## [Editor Report · Decision Letter 2]

26 Nov 2021

Assessing equity of the spatial distribution of primary health care facilities in Fuzhou City, China: A comprehensive method

PONE-D-21-20230R2

Dear Dr. YOU,

We’re pleased to inform you that your manuscript has been judged scientifically suitable for publication and will be formally accepted for publication once it meets all outstanding technical requirements.

Kind regards,

Srinivas Goli, Ph.D.

Academic Editor

PLOS ONE

Additional Editor Comments (optional):

Considering my own reading and reviewers suggestion, I am going with a decision of accept.
---

## [Editor Report · Acceptance letter]

15 Dec 2021

PONE-D-21-20230R2 

Assessing equity of the spatial distribution of primary health care facilities in Fuzhou City, China: A comprehensive method 

Dear Dr. YOU:

I'm pleased to inform you that your manuscript has been deemed suitable for publication in PLOS ONE. Congratulations! Your manuscript is now with our production department. 

Kind regards, 

on behalf of

Dr. Srinivas Goli 

Academic Editor

PLOS ONE